# Research on High Power Laser Damage Resistant Optically Addressable Spatial Light Modulator

Tongyao Du [1,2], Dajie Huang [2], He Cheng [2], Wei Fan [2,3,*], Zhibo Xing [2,3], Jianqiang Zhu [2,3] and Wen Liu [1]

1   Department of Optics and Optical Engineering, University of Science and Technology of China, Hefei 230026, China
2   Key Laboratory on High Power Laser and Physics, Shanghai Institute of Optics and Fine Mechanics, Chinese Academy of Sciences, Shanghai 201800, China
3   Center of Materials Science and Optoelectronics Engineering, University of Chinese Academy of Sciences, Beijing 100049, China
*   Correspondence: fanweil@siom.ac.cn

**Abstract:** Liquid crystal spatial light modulators (LC-SLMs) are devices that can accurately adjust the parameters of beam amplitude, phase, wavefront and polarization. However, due to the limitation of laser damage resistance of component materials, LC-SLMs still have difficulty meeting the application and development needs of a high-average power laser system. Here, we proposed an optically addressable spatial light modulator (OASLM) based on a sapphire substrate. Due to the good thermal conductivity of sapphire, the laser damage resistance of the device was greatly improved. The thermal distribution of OASLM based on the sapphire substrate and the K9 substrate is analyzed by a laser-induced temperature rise model. The experimental results also show the excellent performance of sapphire OASLM under high-power CW laser irradiation, its laser power density is increased from 10 W/cm$^2$ to 75 W/cm$^2$, and the working time is more than 30 min. By bonding sapphire to the other side, the laser power density can be increased to 100 W/cm$^2$, and these are completed without active heat dissipation. This method provides a feasible path for high-average-power SLMs.

**Keywords:** laser damage resistance; optically addressable; spatial light modulators; sapphire substrate





## 1. Introduction

As a programmable device that modulates the spatial distribution of light beams, liquid crystal spatial light modulators (LC-SLMs) have been widely used in the fields of beam spatial shaping [1], pulse shaping [2], adaptive optics [3], beam steering [4] and laser processing [5]. Its application in the field of AR/VR and 6G communication also shows considerable development prospects [6,7]. Due to the absorption characteristics of each layer of materials in the structure of liquid crystal devices [8], most liquid crystal devices now work mainly under low power density and low flux laser. In recent years, with the rapid development of laser fusion, free-space communications, metal 3D printing and other applications, the requirement of damage threshold for spatial light modulators is higher [9–11]. The thermal effect and damage caused by thermal absorption are one of the main problems with liquid crystal optical devices in high-average power laser systems.

Depending on the way that information is written into devices, the SLMs can be divided into electrically addressed (EA) and optically addressed (OA) [12]. At present, the conductive layer in these two SLMs structures is mainly transparent Indium–Tin–Oxide (ITO) conductive film. The laser damage resistance of SLMs is first limited by ITO conductive film [13]. With decades of development, EA has become the main method in commercial SLMs, and there have been many studies on its damage characteristics in the high-power system. Cao et al. reported the performance changes in liquid crystal optical devices irradiated by 808 nm high-power CW laser [14]. When the laser power density

is greater than 133 W/cm$^2$ (the spot diameter on the LC cell is 3 mm), the liquid crystal phase transition occurs, and the modulation characteristics of the device change. Watson et al. studied the phase modulation ability of SLMs irradiated by a 1083 nm high-power CW laser [15]. The experimental results show that when the laser power density is higher than 100 W/cm$^2$ (the device is equipped with active heat dissipation measures), the phase modulation depth of SLMs decreases obviously, which is mainly caused by heat deposition. Zhou et al. used the water-cooled auxiliary heat dissipation system [16], the liquid crystal optical device can resist 11 W laser intensity under 1064 nm CW laser irradiation, and the temperature rise in the center of the device is only 1 K. Gu et al. reported that the conductive layer material with small absorption was selected and the silicon substrate was treated with water cooling [17]. The liquid crystal optical device is irradiated under a CW laser of 1550 nm and 400 W/cm$^2$ for 1 min without damage.

OASLMs are essential for realizing various optical applications that EASLMs are difficult to realize, including coherent to incoherent image conversion, real-time optical correlation as a nonlinear optical medium [18–20]. OASLMs have also been applied in some high-power systems; the US NIF [21], European LMJ [22] and China's "Shenguang" series facilities [23] all use OASLMs for beam shaping and pre-shielding at the injection laser system. Matthews et al. proposed using OASLMs as a photomask [24], demonstrating a metal additive manufacturing method of surface exposure, which greatly improves the speed of additive manufacturing. Xing et al. proposed a high-damage threshold liquid crystal optical switch based on GaN [25], which is expected to be applied to the near-infrared region of the spatial light modulator. Cai et al. also reported that surface-scanning metal additive manufacturing could be realized by using OASLM [26]. Most of these OASLMs work under low power or pulsed laser. With the development of high-power lasers and high repetition rate laser technology, OASLMs also face the problem of thermal deposition under high-average-power lasers. However, there are few reports on the improvement of the laser damage resistance of OASLM under high average power. We investigated the compensation method for the performance degradation of OASLM under high-power laser irradiation, which can increase the laser damage resistance of the device by 2.5 times, but it is far from enough [27]. Sapphire is an excellent multifunctional material with high-temperature resistance, good thermal conductivity, high hardness, optical transparency and excellent chemical stability [28–30]. Due to the excellent performance of sapphire, it has been used as an effective heat dissipation medium [31]. Using sapphire to improve the structure of OASLMs may improve the thermal deposition problem in OASLMs and enhance their laser damage resistance under a high-average-power laser.

In this paper, sapphire was proposed as the liquid crystal cell substrate of OASLM and acted as the heat dissipation component of OASLM. By establishing a laser irradiation model and laser irradiation test, the laser damage resistance of the device under CW laser irradiation was researched. The results show that under the irradiation of high-power continuous light, the temperature rise of the sapphire substrate is smaller than that of traditional K9 glass, especially in the liquid crystal layer. Under the irradiation of 75W/cm$^2$ laser for 30 min, the contrast ratio can still be higher than 100:1. A single sapphire produces a large thermal gradient in the thickness direction of the liquid crystal cell. After the sapphire substrate is bonded on the side of the $Bi_{12}SiO_{20}$ (BSO) crystal, this problem is effectively further improved, and the withstand laser power is increased to 100 W/cm$^2$. The OASLM based on the sapphire substrate provides a feasible path for high-laser-damage-resistant SLMs.

## 2. Simulations and Methods

When OASLM was transmitted in transmission mode, the laser beam propagated through the aperture of the device, as shown in Figure 1. The sapphire substrate and BSO substrate were plated with ITO conductive film. The liquid crystal molecules were filled between the two substrates as the medium for modulating the beam. The liquid crystal molecules were oriented through the polyimide (PI) layer, and the orientation directions of

the substrates on both sides were perpendicular to each other. The voltage was applied to the liquid crystal cell through the wire connected to the ITO layer so that the LC layer and BSO layer formed a series circuit. When the 460 nm addressing beam irradiated on the BSO crystal, the voltage of the liquid crystal layer was changed by using the photoconductive characteristics of BSO to realize beam modulation. The main heat source of OASLM under laser irradiation is the absorption of the laser by ITO film. The heat cannot be transmitted out in time and deposited on the device, resulting in an increase in temperature and the decline of OASLM performance. Sapphire has the characteristics of high hardness, high thermal conductivity and good permeability. Replacing the K9 substrate with a sapphire substrate can increase the heat dissipation efficiency of the device and play a great role in improving the laser damage resistance of OASLM.

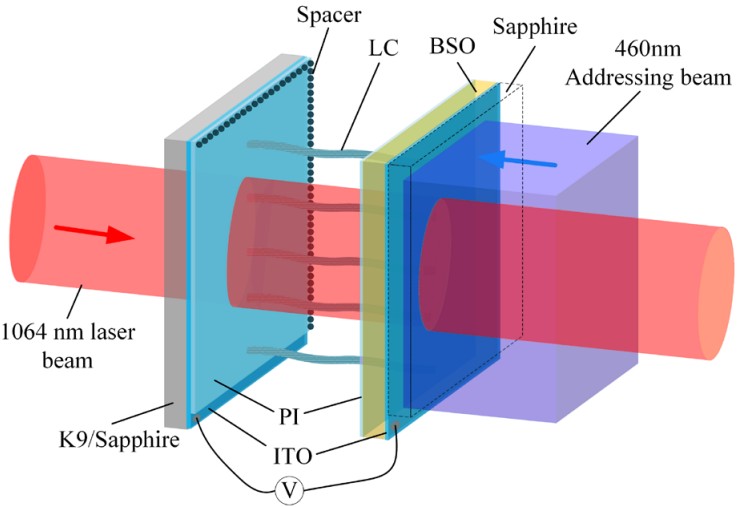

**Figure 1.** OASLM structure illustration. PI: polyimide, ITO: indium tin oxide, LC: liquid crystal, BSO: $Bi_{12}SiO_{20}$ crystal.

In order to analyze the effect of laser induction on OASLM on different substrates, the Beer-Lambert law was used to describe the absorption of the incident laser by materials. Since the temperature would change with space and time, the origin of the cylindrical coordinate system was taken to coincide with the Gaussian beam irradiation center at the center of the sample S surface, and the *z*-axis is the laser irradiation direction, as shown in Figure 2. The internal temperature distribution of the material can be expressed as follows according to the Fourier heat conduction equation:

$$\rho C \frac{\partial T}{\partial t} - \frac{k}{r} \frac{\partial}{\partial r}\left(\frac{\partial T}{\partial r}\right) - k\frac{\partial^2 T}{\partial z^2} = Q = \alpha(T)I \tag{1}$$

where $\rho$, $C$ and $k$ are the density, specific heat capacity and thermal conductivity of the material; $Q$ is the bulk heat source; $\alpha$ is the absorption coefficient of the material to the laser; and $I$ represents the laser intensity of the incident laser.

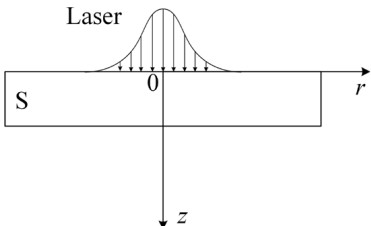

**Figure 2.** Schematic diagram of cylindrical coordinate calculation, S represents the sample irradiated by laser.

This model only considers the temperature rise under CW laser irradiation. For the case of a pulsed laser, the nonlinear effect usually needs to be considered, and this model is not necessarily applicable. Under laser irradiation, the ITO layer is the main heat source in the OASLM, and its absorption coefficient is relatively high. It can be seen from the structure of OASLM in Figure 1 that the position of the ITO layer is inside the liquid crystal layer (ITO1) and outside the BSO crystal (ITO2), respectively. The temperature of ITO1 on the K9 substrate has a great direct impact on the liquid crystal layer. Using sapphire with a higher thermal conductivity as a substrate can make the heat generated by ITO1 easier to dissipate. Based on the structural distribution of OASLM, a laser-induced heat transfer module coupled with multiple physical fields was established. By using the finite element analysis method, through the establishment of the sub-condition domain and the improvement and adjustment of boundary conditions, the temperature distribution of OASLM induced by laser could be analyzed, and the feasibility of the proposed scheme was preliminarily verified. Some parameters of materials used in the model calculation are shown in Table 1. The absorption coefficient was at 1064 nm laser, and the thermal conductivity was at room temperature.

**Table 1.** Some material parameters related to thermal model.

| Parameters | K9 | Sapphire | PI | ITO | LC | BSO |
|---|---|---|---|---|---|---|
| Absorption coefficient ($m^{-1}$) | 0.5 | 0.006 | $10^4$ | $5 \times 10^5$ | 0.1 | 0.2 |
| Thermal conductivity (W/(m·K)) | 1.4 | 34 | 0.28 | 3.3 | 0.5 | 5 |
| Thickness (mm) | 3 | 3 | $10^{-4}$ | $5 \times 10^{-5}$ | $4.8 \times 10^{-3}$ | 1 |

The temperature of OASLM with different substrates was simulated. It was assumed that the intensity of the incident laser is Gaussian distribution, as shown in Figure 2. The OASLM model using K9 substrate and sapphire substrate was solved. The power density of the incident laser was set to 30 W/cm$^2$, which is the Gaussian distribution. The caliber of the K9 and sapphire substrate was set to 35 × 35 mm, and the caliber of the BSO substrate was set to 25 × 25 mm. The redundant part of the substrate was reserved for connecting wires and subsequent thermal management. In order to simplify the calculation, only 1/4 of the model was investigated by using the symmetry characteristics. In the case of temperature rise, the absorption change was ignored. The reflection of the incident interface to the laser was ignored. When the temperature rise of the model is small, the changes caused by temperature on the absorption coefficient and other parameters are ignored. The obtained temperature distribution is shown in Figure 3. Figure 3a,b show the cross-sectional temperature distribution of K9 OASLM and sapphire OASLM. The temperature distribution is gradient distribution due to the Gaussian beam, and the temperature of sapphire OASLM is much lower than that of K9 OASLM. The central temperature of the liquid crystal layer is the main index to evaluate the performance of the equipment, which means that the ITO heat generation and heat dissipation of the liquid crystal layer are the main factors affecting the performance. Figure 3c shows the temperature curves of the substrate surface, the liquid crystal layer, and the ITO layer in the x direction at the cross-section. The embedded diagram shows the structural positions of each layer of the model. It can be seen from Figure 3c that the temperature of the liquid crystal layer is almost the same as that of ITO1, which shows that ITO1 is the main factor affecting the temperature of the liquid crystal layer. After using a sapphire substrate, the heat of ITO1 can be transmitted in time, and the temperature of the liquid crystal layer can be maintained at a good level compared with that of K9. Figure 3d shows that the thicker the sapphire substrate is, the more obvious the thermal diffusion effect is, while the improvement effect is gradually reduced. Therefore, a 3 mm thickness was adopted in the subsequent analysis.

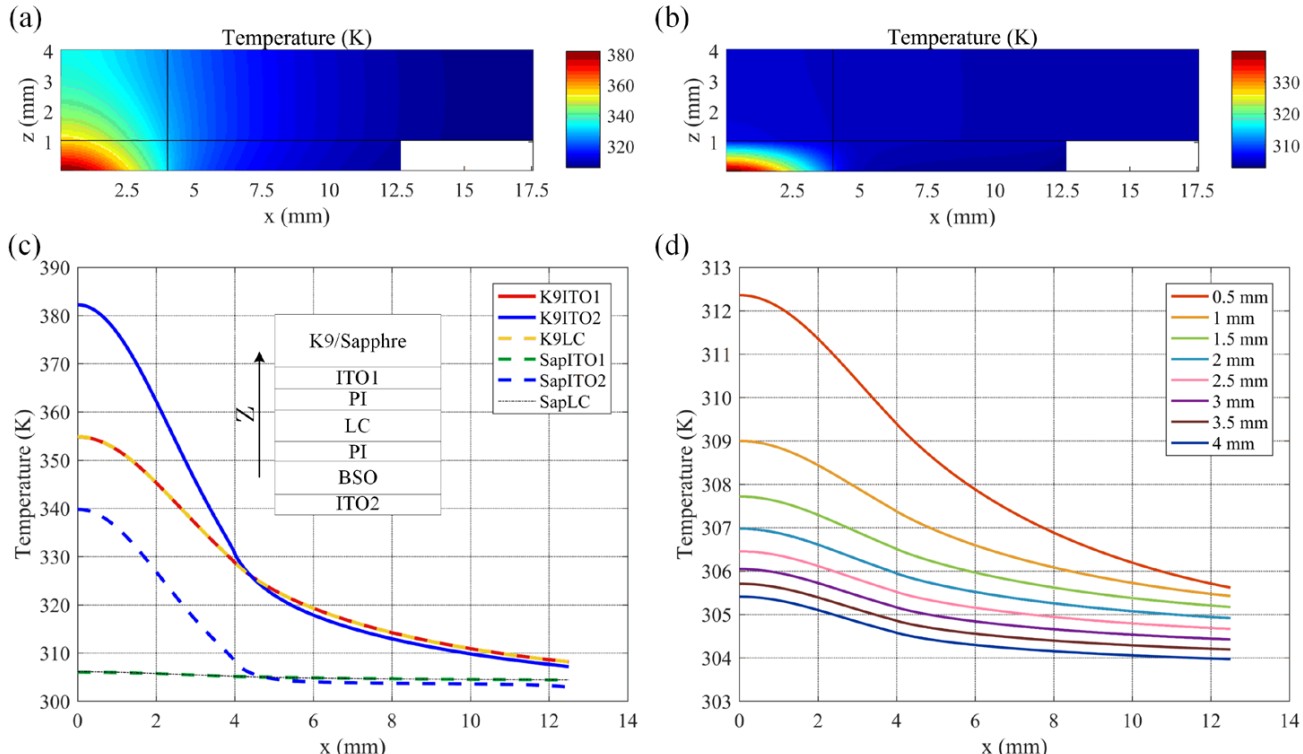

**Figure 3.** (**a**) Temperature distribution of K9 substrate OASLM. (**b**) Temperature distribution of Sapphire substrate OASLM. (**c**) Temperature curve along the *x*-axis of each layer structure. The embedded diagram represents the structural position of each floor in the model. (**d**) The influence of sapphire with different thicknesses on the temperature of liquid crystal layer.

However, the results also show that the temperature of ITO2 on the BSO side is high; although the temperature decreases after using a sapphire substrate, there is still a large temperature gradient in the thickness direction of the liquid crystal cell. This is because the thermal conductivity of BSO crystal itself is very low, and ITO is difficult to dissipate heat in direct contact with air, resulting in most of the heat deposited on BSO crystal. Additional negative effects may be introduced under high-power laser irradiation, affecting the normal operation of OASLM equipment. It may be assumed that a layer of the sapphire substrate is bonded on the BSO side as the heat-dissipation substrate. The improved bilateral sapphire model is shown in Figure 4a; sapphire and ITO2 fit perfectly in the model. Figure 4b shows the temperature distribution of bilateral sapphire OASLM. The results show that the heat of ITO on the BSO side is transmitted from sapphire, and the center temperature decreases significantly. Figure 4c shows the temperature curve of the main layer; the temperature difference in the thickness direction does not exceed 3 K, which improves the previous problem. In addition, it is worth noting that the heat of the whole device is mainly concentrated in the liquid crystal layer because, in addition to ITO, the PI layer also has a certain absorption effect, but it is much smaller than that of ITO. The sapphire on the BSO side is completely attached to the ITO in the model, so the heat conduction effect is obvious. In the actual equipment, it is difficult to achieve such a high degree of fit, so there is some loss of the final heat dissipation effect.

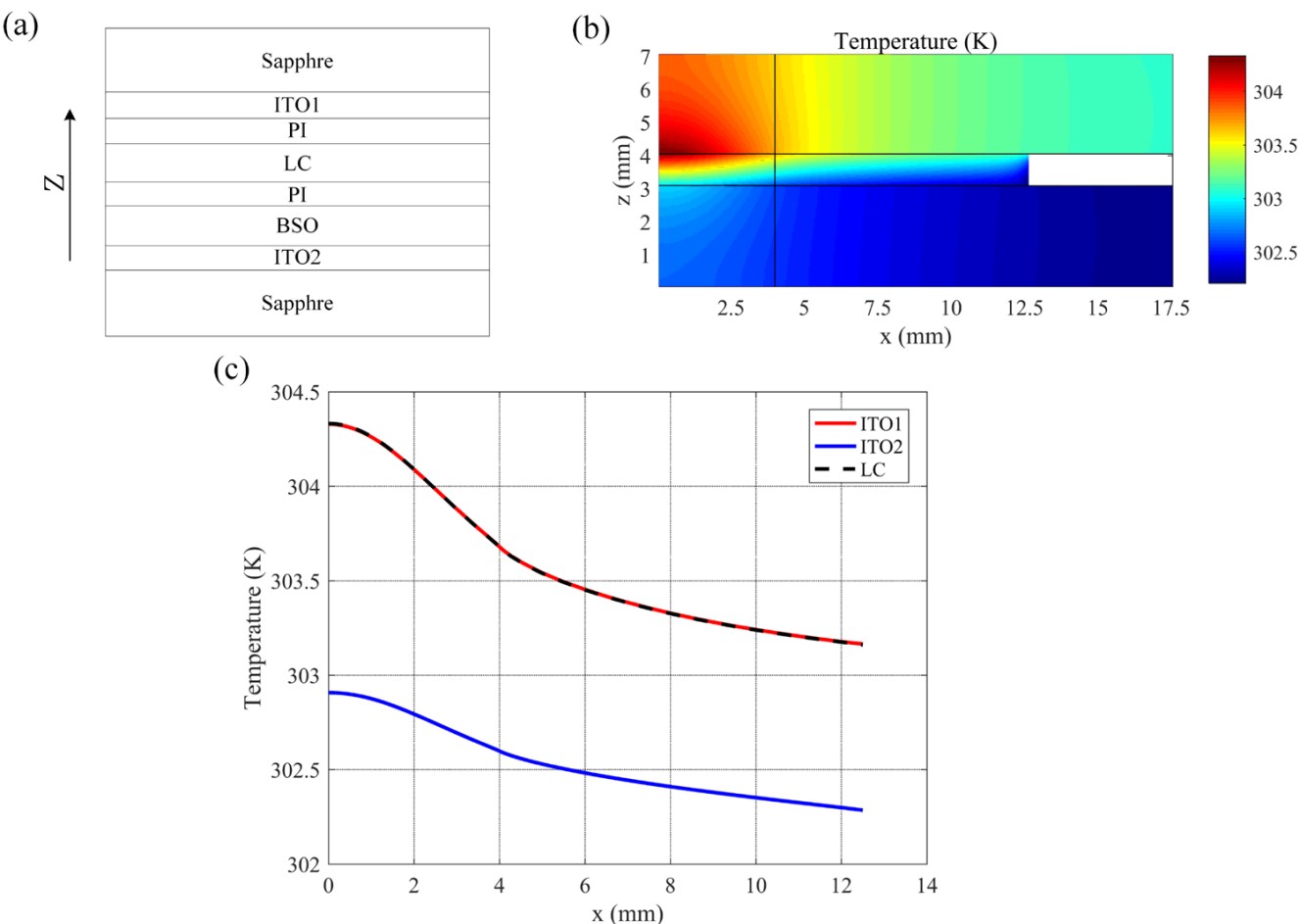

**Figure 4.** (**a**) Schematic diagram of bilateral sapphire OASLM structure. (**b**) Temperature distribution of bilateral sapphire OASLM. (**c**) Temperature curve along the *x*-axis of each layer structure.

## 3. Experimental Results and Discussion

The experimental device for the laser damage resistance test is shown in Figure 5. The laser adopted a power-adjustable fiber CW laser with a maximum output power of 500W, and the central wavelength was 1064 nm. The sample to be measured was placed in the dotted box, and the spot diameter on the sample was 8 mm. The thermal imager is FLK-Ti400+, produced by Fluke. Because the temperature distribution inside the liquid crystal layer cannot be measured directly in the experiment, the temperature distribution on the surface of the irradiated substrate was detected by a thermal imager. The OASLM used was an amplitude operation mode, so two parallel polarizers were set in the front and back, and the polarization direction was perpendicular to the horizontal line. The optical aperture of OASLM equipment is 22 × 22 mm, with 720 × 720 pixels. They were calibrated through the calibration program [32]. The switching ratio under low-power laser irradiation was greater than 100:1. The light field distribution of laser passing through OASLM was detected by the CCD camera, and the performance of recording equipment changed with the increase in laser power. The records under different laser intensities are independent. The power detector records the laser intensity incident on the OASLM through sampling.

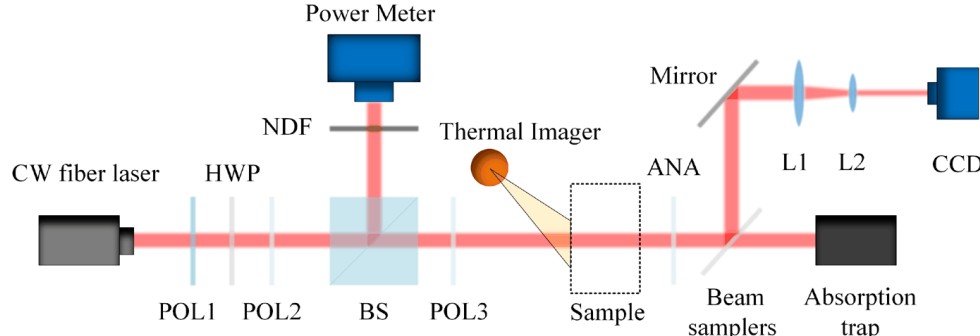

**Figure 5.** CW laser damage resistance test system. POL: polarizer, HWP: half wave plate, BS: beam splitter, NDF: neutral density filter, ANA: analyzer, L: lens.

The temperature of the liquid crystal layer was directly affected by the heat generation of ITO on the K9 or sapphire substrate. Firstly, the temperature rise of the K9 substrate and sapphire substrate coated with ITO film was tested. The irradiation time was generally more than 30 min to ensure the stability of the temperature of the equipment. Figure 6 shows the temperature response of each substrate under irradiation of different power densities, where the temperature value is the highest surface temperature on the ITO side. The results show that the K9 substrate reached 315 K at the power of 10 W/cm$^2$ and increased rapidly with the increase in laser power. It reached 336 K at 37 W/cm$^2$, close to the clear point of liquid crystal material. The temperature rise of the sapphire substrate was relatively mild, and the sapphire substrate with a thickness of 3 mm was only 312 K under 100 W/cm$^2$ laser irradiation. The temperature response of sapphire with a thickness of 1 mm and 3 mm showed that the substrate with a large thickness had a better heat dissipation effect. The thicker substrate was easier to meet the process requirements in the production of liquid crystal cells, and the too-thin substrate was needed to control the stress to prevent deformation. Therefore, the sapphire substrate with a thickness of 3 mm was used to make the liquid crystal cell of OASLM.

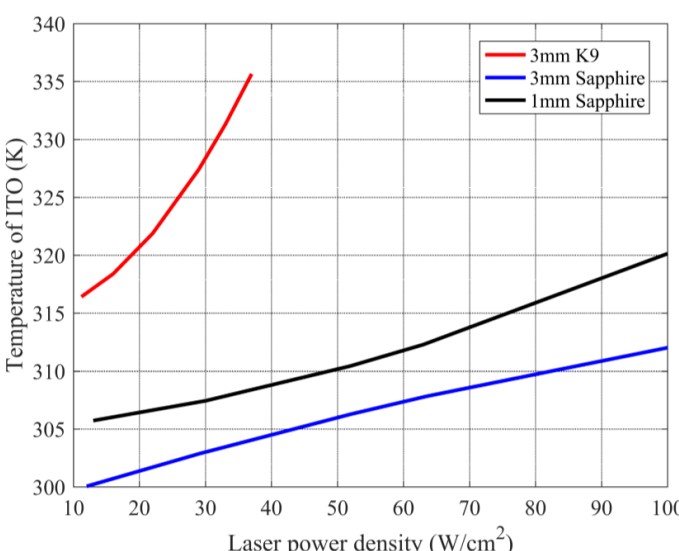

**Figure 6.** The temperature of ITO tested on different substrates varies with laser power density.

The basic optical properties of sapphire OASLM were analyzed. Sapphire OASLM was placed in the optical path of a low-power laser for switching mode and pattern mode response test. In OASLM, the BSO crystal and liquid crystal layer were in a series structure, and the voltage loaded on the liquid crystal cell device was adjusted by driving voltage. Then, the driving current was used to adjust the light intensity of the addressing beam.

The change in the light intensity of the addressing beam changes the resistance of the BSO crystal and the voltage of the liquid crystal layer. The light field distribution shown in Figure 7 is recorded in different modes. Figure 7a is the light field distribution in the off mode, and the light leakage part is the sealant area, which is not located in the light-passing aperture. In the off mode, there is no beam passing through the OASLM aperture. Figure 7b shows the light field distribution in the on mode. Ignoring the noise introduced by the optical element, the light field distribution is uniform, indicating that the overall transmittance of the device has good uniformity. Figure 7c shows the images written into the photomask and output by OASLM, which are logos and grayscale images, respectively. The images are clearly visible and have obvious contrast, meeting the expected requirements. Figure 7d shows that the transmittance of the device changes with the gray value. The highest transmittance can reach 87.5%, and the gamma curve is in line with the expectation.

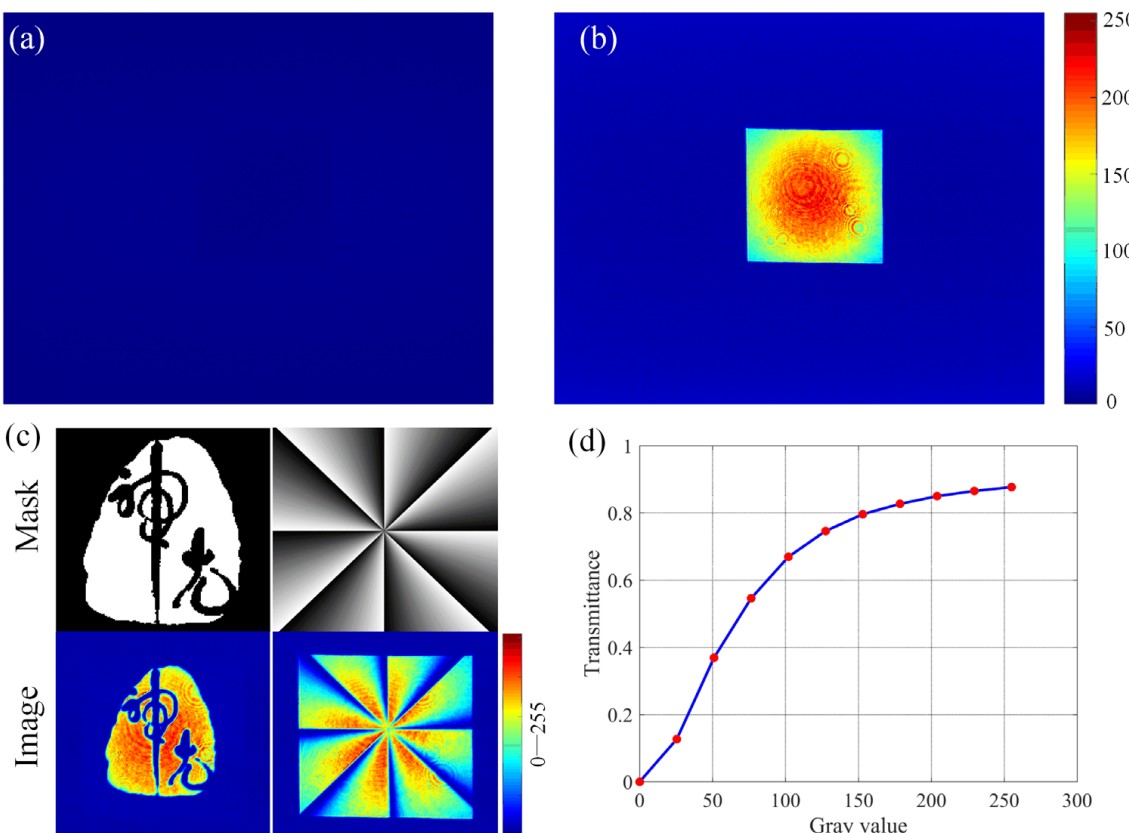

**Figure 7.** Images output by OASLM in each mode. (**a**) OASLM is in off mode. (**b**) OASLM is in on mode. (**c**) Images under different masks, logos and grayscale. (**d**) The transmittance of OASLM corresponds to the gamma curve of the gray value.

After ensuring that the optical performance of sapphire OASLM meets the standard, it is placed in the high-power CW laser system. The initial state of OASLM is in off mode, which means that the CCD camera cannot detect the light intensity. The output of the laser was adjusted to irradiate the OASLM device; under the irradiation of a high-power laser, the temperature affects the liquid crystal parameters, resulting in light leakage in the off mode. The laser damage resistance of OASLM can be judged by the light leakage intensity in the off mode and the actual contrast ratio that the OASLM can achieve. Because the laser is Gaussian distribution, the temperature at the center of the spot is usually the highest, and the light leakage occurs first at this position, so the center of the laser spot was taken as the investigation point. The variation in the maximum temperature on the surface of the liquid crystal cell glass substrate with the power density is shown in Figure 8a. The temperature

rise of sapphire OASLM is very mild compared with K9 OASLM, which reflects that the temperature rise of the liquid crystal layer is also slow. K9 OASLM at the power density of 22 W/cm$^2$, the liquid crystal in the center of the spot reached the clear point; the device completely lost its function. The maximum temperature of sapphire OASLM at the sapphire side is only 309 K at the power density of 75 W/cm$^2$. Due to the low thermal conductivity of BSO crystal, the temperature rises of ITO plated on it were not alleviated. With the increase in laser intensity, the contrast ratio of the device continued to decline. When it reaches 120 W/cm$^2$, the contrast ratio is reduced to the point where the modulation capability is completely lost. The image corresponding to the fringe mask is shown in Figure 8b; the contrast is obvious, which can meet the needs of many high-power laser systems. Figure 8c,d show typical thermal camera images recorded by OASLM under laser irradiation, demonstrating the excellent cooling efficiency of sapphire OASLM. Figure 8c shows that under the irradiation of 10 W/cm$^2$ laser intensity, the maximum temperature of the K9-OASLM center reached 309 K, and the temperature gradient was large. Under the laser intensity of 75 W/cm$^2$, the center temperature of sapphire-OASLM was only about 310 K. Due to its excellent cooling efficiency, the temperature gradient of the device also decreased, as shown in Figure 8d.

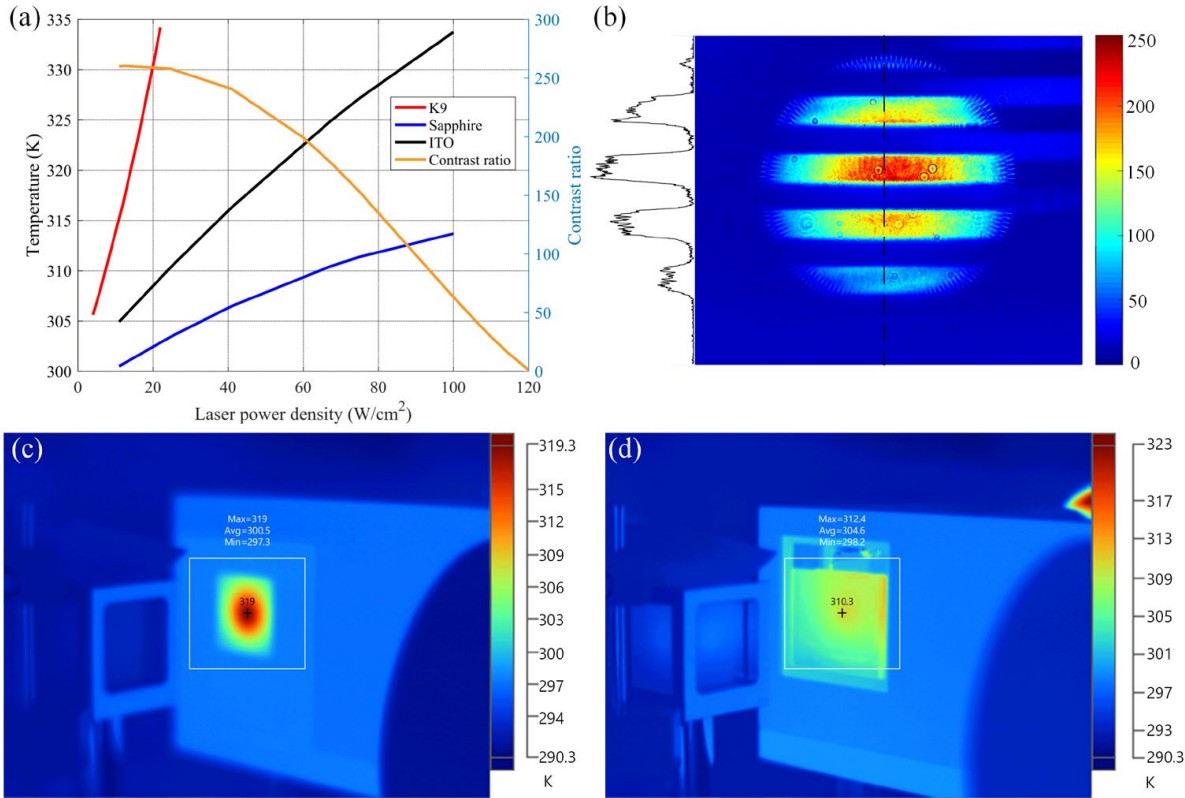

**Figure 8.** (**a**) The substrate surface temperature of K9 and sapphire OASLM device as a function of laser power density and the contrast ratio of sapphire OASLM changes with power intensity. (**b**) Light field distribution of sapphire OASLM output at 75 W/cm$^2$. (**c**) Thermal image of OASLM based on K9 substrate at 10 W/cm$^2$ laser intensity. (**d**) Thermal image of OASLM based on sapphire substrate at 75 W/cm$^2$ laser intensity.

Although the temperature rise on the sapphire substrate side is low, the temperature of the BSO side in sapphire OASLM is not ideal, as shown in Figure 8a. Most of the heat is concentrated in the spot irradiation area, resulting in a large temperature difference between the substrates on both sides. The results of bilateral sapphire in the model show that this method is very effective. Therefore, the sapphire with a thickness of 3 mm was bonded on the side of the BSO substrate by liquid capillary construction method. By using

liquid surface tension, different wafers can be very closely bonded and strongly bonded without pressure and heat treatment [33]. The bilateral sapphire OASLM is placed in the high-power CW laser system, and the temperature change is shown in Figure 9a. The results show that the structure of bilateral sapphire effectively alleviates the thermal deposition on the BSO side, and the temperature of the sapphire substrate side also decreases. From the change in contrast ratio with power intensity, the contrast ratio of the device is reduced to the point that it can no longer work normally at about 155 W/cm². Figure 9b shows the image obtained when the fringe mask is loaded at 100 W/cm². The contrast ratio of the device is 170:1, which can be used effectively. However, due to the immature bonding process between BSO and sapphire, it is difficult to achieve the perfect bonded between the two surfaces; only a certain degree of gluing was achieved. This may limit the heat dissipation effect of sapphire and may also affect the optical performance of the device. If a more mature process is adopted for treatment, the power density can be further improved, and the optical properties can also be improved.

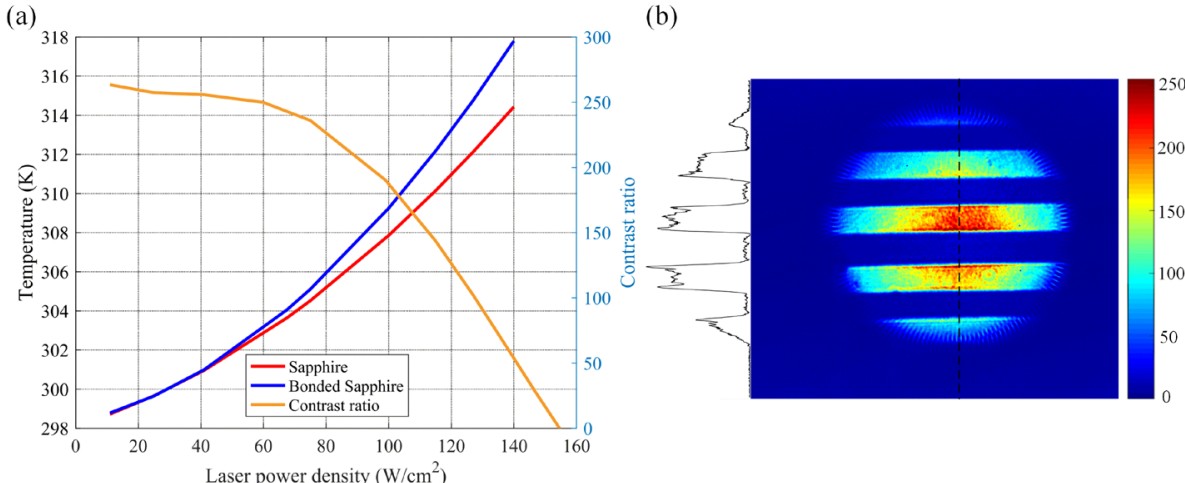

**Figure 9.** (**a**) Temperature of sapphire on both sides of the OASLM device as a function of laser power density and the changes in contrast ratio with power intensity. (**b**) Light field distribution of bilateral sapphire OASLM output at 100 W/cm².

## 4. Conclusions

We demonstrated a high-laser-damage-resistant OASLM based on a sapphire substrate. Compared with the traditional K9 substrate, the heat generated by ITO can be effectively dissipated by replacing the sapphire substrate. Firstly, the effectiveness of the sapphire substrate was analyzed by a laser-induced model. Under the same laser power irradiation, the maximum temperature of the sapphire OASLM liquid crystal layer was 20 K lower than that of K9 OASLM, which is an obvious advantage. Then, the sapphire and K9 plated with ITO were simply tested. The results showed that the sapphire with 3 mm thickness is the substrate with the best heat dissipation effect. The optical performance index of sapphire OASLM also reached the expectation, and the power density can reach 75 W/cm² under the 8mm diameter spot, which is obtained when the working time is more than 30 min. A sapphire substrate was bonded on the BSO side, which improves the power density of OASLM to 100 W/cm². However, the bonding process is not mature, which does not achieve the best effect. It is worth noting that the work in this paper was achieved under the condition of passive heat dissipation. If further water cooling and other heat management are carried out, the laser damage resistance can be greatly increased again. The method proposed in this paper can make OASLM more widely used in a high-power laser system, and the method using sapphire substrate can also be applied to other types of SLMs, which provides a feasible path for the development of high-laser-damage-resistant SLMs.

**Author Contributions:** D.H., H.C., W.F., J.Z. and W.L. contributed to the conception and design of the study. T.D., H.C. and Z.X. organized the database. T.D. performed the statistical analysis. T.D. wrote the first draft of the manuscript. D.H. and W.F. wrote sections of the manuscript. All authors have read and agreed to the published version of the manuscript.

**Funding:** This work was supported by the Strategic Priority Research Program of the Chinese Academy of Sciences, China (Grant Nos. XDA25020303). Innovation Promotion Association of the Chinese Academy of Sciences (2022243).

**Institutional Review Board Statement:** Not applicable.

**Informed Consent Statement:** Not applicable.

**Data Availability Statement:** The data included in this study are all owned by the research group and will not be transmitted.

**Conflicts of Interest:** The authors declare no conflict of interest.

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
