# Peer review of "Research on High Power Laser Damage Resistant Optically Addressable Spatial Light Modulator"

_photonics, doi:10.3390/photonics9110811_

Round 1
Reviewer 1 Report
The manuscript by Du et al. presents the research on high power laser damage resistant optically addressable spatial light modulator. The subject is presented clearly, with detailed discussions regarding the various experimental solutions to achieving the high laser damage resistant OASLM based on sapphire substrate in various configurations. Moreover, the manuscript provides a feasible path for the development of high laser damage resistant SLMs.
The manuscript is correctly presented, although the quality of the English could be improved in places, and the technical details are well-covered and the figures are adequate. However, it is worth noting that punctuation error and abbreviation problem in the manuscript.
I would be happy to recommend the work for publications in photonics after these concerns have been addressed.
Author Response
Thank you for your comments concerning our manuscript. Those comments are all valuable and very helpful for revising and improving our paper, as well as the important guiding significance to our research. We have studied comments carefully and have made correction which we hope meet with approval.

Author Response

(The authors gave the same response as above.)

Reviewer 3 Report
Please see the attached file.

Author Response

(The authors gave the same response as above.)
